# Low Selectivity Indices of Ivermectin and Macrocyclic Lactones on SARS-CoV-2 Replication In Vitro

Christine Chable-Bessia [1,†], Charlotte Boullé [2,3,†], Aymeric Neyret [1], Jitendriya Swain [4], Mathilde Hénaut [1], Peggy Merida [4], Nathalie Gros [1], Alain Makinson [2,3], Sébastien Lyonnais [1], Cédric Chesnais [2,3,*] and Delphine Muriaux [1,4,*]

1   Centre d'Etude des Maladies Infectieuses et Pharmacologie Anti-Infectieuse (CEMIPAI), CNRS UAR 3725, Université de Montpellier, 1919 Route de Mende, CEDEX 05, 34293 Montpellier, France; christine.chable-bessia@cemipai.cnrs.fr (C.C.-B.); aymeric.neyret@cemipai.cnrs.fr (A.N.); mathildehenaut@outlook.fr (M.H.); nathalie.gros@cemipai.cnrs.fr (N.G.); sebastien.lyonnais@cemipai.cnrs.fr (S.L.)
2   TransVIHMI, Institut de Recherche pour le Développement (IRD), Unité Mixte Internationale 233, INSERM Unité 1175, Université de Montpellier, 34293 Montpellier, France; boulle.charlotte@gmail.com (C.B.); a-makinson@chu-montpellier.fr (A.M.)
3   Infectious Disease Department, University Hospital of Montpellier, 34293 Montpellier, France
4   Institute of Research in Infectiology of Montpellier (IRIM), University of Montpellier, UMR9004 CNRS, 34293 Montpellier, France; jitendriya.swain@irim.cnrs.fr (J.S.); peggy.merida@irim.cnrs.fr (P.M.)
*   Correspondence: cedric.chesnais@ird.fr (C.C.); delphine.muriaux@cemipai.cnrs.fr (D.M.)
†   These authors contributed equally to this work.

**Abstract:** Ivermectin was first approved for human use as an endectocide in the 1980s. It remains one of the most important global health medicines in history and has recently been shown to exert in vitro activity against SARS-CoV-2. However, the macrocyclic lactone family of compounds has not previously been evaluated for activity against SARS-CoV-2. The present study aims at comparing their anti-viral activity in relevant human pulmonary cell lines in vitro. Here, in vitro antiviral activity of the avermectins (ivermectin and selamectin) and milbemycins (moxidectin and milbemycin oxime) were assessed against a clinical isolate from a CHU Montpellier patient infected with SARS-CoV-2 in 2020. Ivermectin, like the other macrocyclic lactones moxidectin, milbemycin oxime and selamectin, reduced SARS-CoV-2 replication in vitro (EC50 of 2–5 μM). Immunofluorescence assays with ivermectin and moxidectin showed a reduction in the number of infected and polynuclear cells, suggesting a drug action on viral cell fusion. However, cellular toxicity of the avermectins and milbemycins during infection showed a very low selectivity index of <10. Thus, none of these agents appears suitable for human use for its anti-SARS-CoV-2 activity per se, due to low selectivity index.

**Keywords:** SARS-CoV-2; in vitro drug screening; ivermectin; anti-parasite drugs; antiviral specificity index

## 1. Introduction

In December 2019, severe acute respiratory syndrome coronavirus 2 (SARS-CoV-2) emerged in Wuhan, China [1,2]. SARS-CoV-2 is a *betacoronavirus*, which are enveloped viruses containing single-strand, positive-sense RNA. No effective treatment or treatment regimen has been established for coronavirus disease-2019 (COVID-19) to date and although success has been achieved with vaccination modalities, there remains a need for therapeutic approaches for the unvaccinated and for vaccine escape mutants. Screening existing, regulatory approved medicines for potential activity against SARS-CoV-2 offers the potential to reduce unnecessary animal and clinical evaluations. Repurposing requires a comprehensive understanding of the pharmacokinetic/pharmacodynamic relationship of the product and changes from the approved posology result in potentially significant data generation requirements. Several candidates have emerged from this strategy, although

a convincing antiviral therapy has yet to emerge. The directly antiviral remdesivir has conflicting efficacy data, and lopinavir/ritonavir, hydroxychloroquine and ivermectin have been successively ruled out in randomized clinical trials [3–6]. Dexamethasone for oxygen-requiring patients has evidence of clinically relevant efficacy in the RECOVERY trial [7], through a reduction of the inflammatory response in the latter phase of the infection. Therefore, the therapeutic arsenal is still limited and further options are needed [8].

Ivermectin (IVM) is a widely used antiparasitic drug targeting the gamma aminobutyric acid and glutamate-gated chlorine channel in invertebrates, causing flaccid paralysis of parasites and becoming one of the world's most important global heath medicines, a discovery awarded a Nobel Prize in 2015. Apart from its antiparasitic effect against filariae and ectoparasites, there is evidence of anti-inflammatory and antiviral properties on RNA viruses, mostly of the flaviviridae family (including dengue (DNV), zika, yellow fever) [9]. A study by Caly et al. [10] reported that IVM inhibited SARS-CoV-2 in vitro using the Vero-hSLAM assay. However, the concentration resulting in 50% inhibition (IC50, 2 to 2.5 µM, i.e., 1750 to 2190 ng/mL) was several times higher than the maximal plasma concentration ($C_{max}$) that can be reached after the oral administration of the FDA-approved dose for onchocerciasis (200 µg/kg, e.g., 14 mg for 70 kg) [11–13]. Human lung exposure is thought to reach a maximum of 772 ng/mL [14]. IVM doses up to 120 mg have been administered and appeared well tolerated, but with limited experience; the in vitro IC50 by Caly et al. was >9-fold and >21-fold higher than the day 3 plasma and lung tissue simulated $C_{max}$, respectively, following a higher dose IV regimen of 600 µg/kg dose daily for 3 days. This dose scenario exceeds the highest regulatory approved dose of IVM of 200 µg/kg single dose for the treatment of strongyloidiasis.

IVM is generally well tolerated [15], except for potential neurotoxicity that could be linked to GABA-gated channel in the central nervous system of mammals if the blood–brain barrier (BBB) integrity is damaged, a possible mechanism of COVID-19 encephalopathy during the cytokine storm. This neurotoxicity in enhanced in the case of mutations of the P-glycoprotein (P-gp), one of the main BBB efflux pumps, for which IVM is a substrate.

Put together, these results highlight the importance of pharmacokinetic considerations, acknowledging that discrepancies between in vitro and in vivo results have been observed. For instance, DNV's (dengue virus) IC50 is even higher (17 µM) [16].

SARS-CoV-2 enters host cells by binding to angiotensin-converting enzyme 2 (ACE2) receptors, widely distributed in human tissue but predominantly in human alveolar epithelial cells [17,18]. Mechanisms underlying the possible antiviral effect of IVM include its ability to bind to the host's importin (IMP) $\alpha$, leading to dissociation of the IMP $\alpha/\beta 1$ heterodimer and modifying the interaction of IMP with viral proteins therefore hampering nuclear import [19,20]. Other hypothesis include allosteric modulation of the $\alpha 7$ nicotinic acetylcholine receptor (nAChR) [21,22].

IVM belongs to the wider family of macrocyclic lactones, being synthesized through a semi-fermentation pathway through bacteria of the *Streptomyces* genus. The macrocyclic lactones are divided into two classes: avermectins (ivermectin, doramectin, selamectin) and milbemycins (moxidectin and milbemycin oxime) depending on the substitution at the position 13 of the macrolide ring, also common to macrolide antibiotics (also investigated as potential antiviral molecules in SARS-CoV-2 infection). IVM and moxidectin (MOX, achieved FDA approval for onchocerciasis [23–25]) are the only macrocyclic lactones approved for human use by a regulatory authority, while selamectin, doramectin and milbemycin oxime are approved for veterinary use only. Similarities across these drugs include mode of action and lipophilic properties leading to preferential deposition in adipose tissues, but they differ in the degree they are substrates for transporters, GABA and glutamate gated chloride channel binding, and their respective absorption, distribution, metabolism and excretion (ADME) profiles. Moxidectin's human elimination half time ($t_{1/2}$) reaches >20 days following oral administration [26,27]. MOX is also a weak substrate of P-gp, and has demonstrated lower neurotoxicity in models of neurotoxicosis.

To explore the potential of this class of compounds against SARS-CoV-2, we screened avermectins and milbemycins for their antiviral efficacy in vitro.

## 2. Materials and Methods

### 2.1. Cell Lines and Culture

Vero E6 cells (African green monkey kidney cells) were obtained from ECACC (Sigma-Aldrich, Merck, Darmstadt, Germany) and maintained in Dulbecco's minimal essential medium (DMEM) supplemented with 10% heat inactivated fetal bovine serum (FBS, Thermo Fisher, Waltham, MA, USA) 50 U/mL of penicillin (Ozyme, Saint-Cyr-l'école, France), 50 μg/mL of streptomycin (Ozyme, France) and 25 mM of HEPES at 37 °C with 5% $CO_2$. The human pulmonary alveolar A549-hACE2 cells were engineered using a lentiviral vector from Flash Therapeutics (Toulouse, France) to express the human receptor ACE2. A549 cells obtained from ECACC (#86012804, Sigma-Aldrich, Merck, Darmstadt, Germany) were transduced and maintained in RPMI (Ozyme, France) supplemented with 10% heat inactivated fetal bovine serum (FBS), 50 U/mL of penicillin, 50 μg/mL of streptomycin, 25 mM of HEPES and 1 mM of sodium pyruvate at 37 °C with 5% $CO_2$. Cells were then sorted by FACS in order to obtain an A549-hACE2 population stably expressing hACE2. Calu-3 cells (epithelial lung adenocarcinoma cells) were obtained from Elabscience (#EP-CL-0054, Tebu-Bio, Le Perray-en-Yvelines, France) and maintained in DMEM supplemented with 10% heat-inactivated FBS, 50 U/mL of penicillin, 50 μg/mL of streptomycin and 25 mM of HEPES at 37 °C with 5% $CO_2$.

### 2.2. SARS-CoV-2 Virus

SARS-CoV-2/CHU Montpellier/France was isolated from CPP Ile de France III, n°2020-A00935−34 and "Centre de Ressources Biologiques" collection of the University Hospital of Montpellier (France). The virus was propagated in Vero E6 cells with DMEM containing 2.5% FBS and 25 mM HEPES at 37 °C with 5% $CO_2$ and was harvested 72 h post inoculation. Virus stocks were stored at −80 °C. All work with infectious SARS-CoV-2 was performed in an approved biosafety level 3 (BSL3) facility by trained personnel at the CEMIPAI UAR3725.

### 2.3. Virus Titration

Virus titration from infected cell culture supernatant was monitored using plaque assays on a monolayer of Vero E6 cells, as previously described [28]. Briefly, Vero E6 cells were inoculated with 10-fold serial dilutions ($10^{-1}$, $10^{-2}$, $10^{-3}$, $10^{-4}$, $10^{-5}$) of a SARS-CoV-2 stock and incubated for one hour at 37 °C with regular rocking. Inoculum was removed and replaced with 500 μL/well of agar overlay (MEM; 0.1% sodium bicarbonate, 0.2% BSA, 20 mM HEPES, 0.6% oxoid agar) (Fisher Scientific, France). Once the overlay had solidified, plates were incubated at 37 °C for 72 h prior to fixing with 4% PFA (Euromedex, Souffelweyersheim, France) and staining with crystal violet (Sigma, France) to visualize plaques. Plaques were quantified and viral titer of the stock sample was determined as plaque-forming units (PFU)/mL by taking the average number of plaques for a dilution and the inverse of the total dilution factor.

### 2.4. Assessment of Antiviral Activity

Drugs were purchased from Sigma Aldrich (France): moxidectin (#Y0000772), doramectin (#33993), milbemycin oxime (#Y0001893), selamectin (#Y0000814), ivermectin (#I8000010). Drug structures are presented in Figure 1a. Remdesivir (#282T7766) was purchase from Tebu (France). Drugs were prepared at a concentration of 10 mM in DMSO (Sigma, France). Antiviral activity was assessed by a cytopathic effect (CPE) reduction assay on Vero E6 cells or by RT-qPCR on A549-hACE2 or Calu-3 cells. Briefly, 30,000 cells per well were cultured in 96-well culture plates (Dutscher, BERNOLSHEIM, France) for 24 h. Drugs were diluted from stock to 50 μM in DMEM or 25 μM RPMI for testing antiviral activity on Vero E6 cells or A549-hACE2 cells, respectively. Cells were incubated with

100 µL of the compounds diluted in DMEM/0.5% DMSO (*v/v*) (Vero E6) or RPMI/0.25% DMSO (A549hACE2) or DMEM/0.25% DMSO (Calu-3) at the indicated concentrations and the plate were incubated for 2 h. Subsequently, cells either were mock infected (for analysis of cytotoxicity of the compound) or were infected with 400 PFU of virus per well (MOI of 0.01) in a total volume of 110 µL of medium with compound. In that context, the experiments were performed in an on-going viral replication in the treated cells.

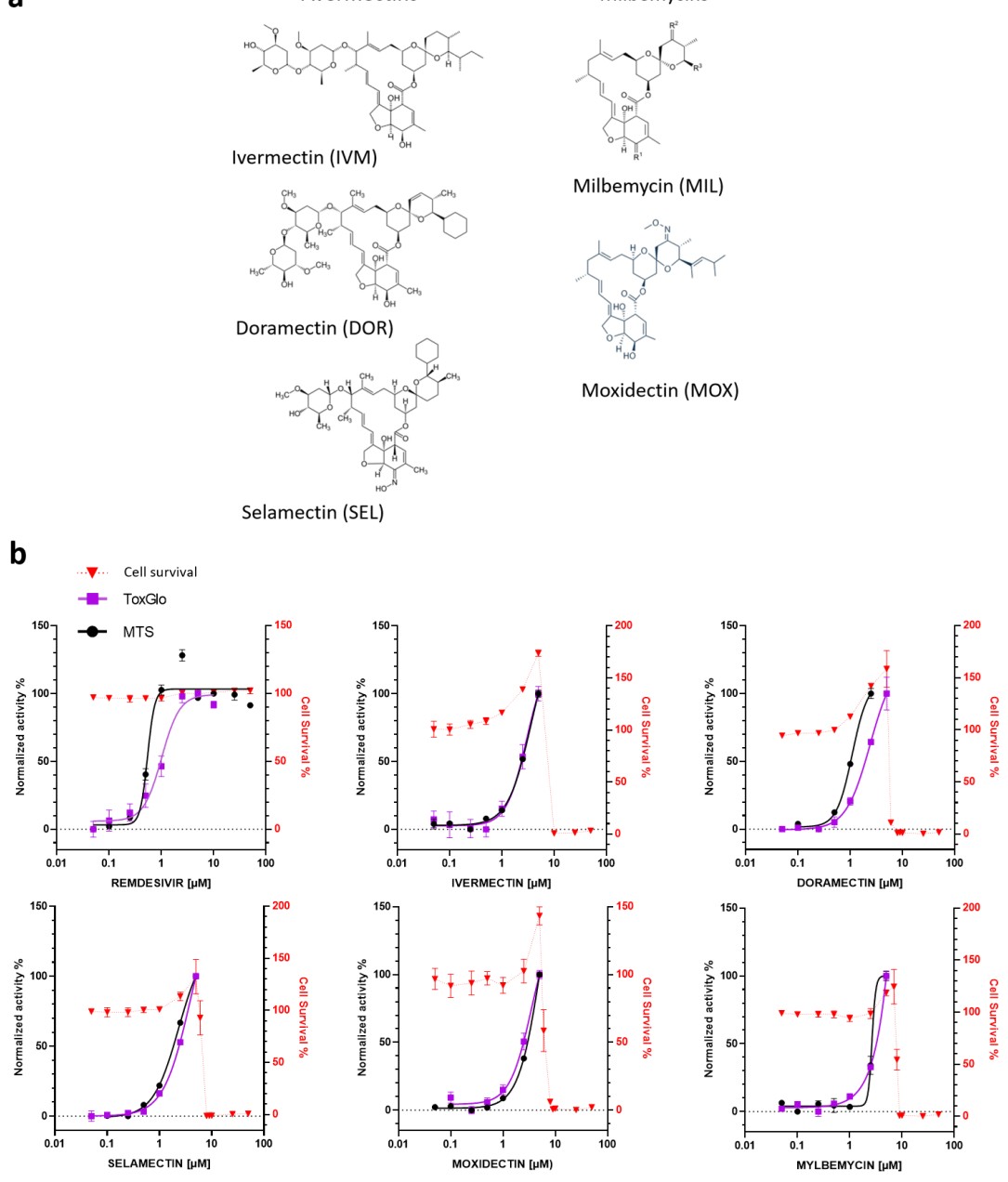

**Figure 1.** Antiviral and cytotoxic activities of selected compounds against SARS-CoV-2 infection in Vero E6 cells. (**a**) Structure of avermectins and milbemycins drugs used in this study. (**b**) Vero E6 cells were infected with SARS-CoV-2 at a MOI of 0.01 with 0.05, 0.1, 0.25, 0.5, 1, 2.5, 5, 10 and 50 µM of the indicated compounds and incubated for 72 h. The cytotoxicity of the drugs to Vero E6 cells was measured by MTS assay. Inhibition of virus replication was either quantified by MTS (black) or ToxGlo (purple) assays. The left and right *Y*-axes of the graphs stand for mean % inhibition vs. DMSO control and mean % cell survival (red), respectively. Data points are mean values and bars are the standard deviation of experiments performed in duplicate.

Cell viability was assessed 3 days post-infection by MTS assay using a Cell Titer 96 aqueous cell proliferation kit (Promega, Charbonières-les-Bains, France) or Viral ToxGlo™ (ToxGlo assay, Promega, France). Absorption at 495 nm (MTS) or luminescence (ToxGlow) were measured with an EnVision multilabel plate reader (PerkinElmer, Villebon S/Yvette, France). The 50% effective concentration (EC50, the concentration required to inhibit virus-induced cell death by 50%) and the 50% cytotoxic concentration (CC50, the concentration that reduces the viability of uninfected cells to 50% of that of untreated control cells) were determined using 4-parameter nonlinear regression with GraphPad Prism v8.0, based on the following calculations:

$$\text{Cell cytotoxicity, \%TOX} = \left(1 - \frac{drug}{cell\ DMSO}\right) \times 100 \tag{1}$$

$$\text{Percentage of viral replication inhibition, \%Inhibit} = \left(\frac{drug - infected\ cells}{cell\ DMSO - infected\ cells}\right) \times 100 \tag{2}$$

Data are mean ± SD of three biological replicates, each of which consisted of duplicate samples. Data were normalized as the relative efficiency or cell viability of inhibitor-treated cells compared with those of untreated cells (set to 100%).

*2.5. Quantitative Reverse Transcription Polymerase Chain Reaction (RT-qPCR)*

Viral particles were collected from virus-infected A549 hACE2 cell supernatants at 72 h post-infection (100 µL) and viral RNAs were isolated from the supernatants using the Quick RNA Viral 96 Kit (Zymo Research, Ozyme, France) according to the manufacturer's instructions. Viral RNAs were isolated from Calu-3 infected cells 48 h post-infection using the Luna Cell Ready Lysis Module (New England Biolabs, UK). Viral RNA were quantified by qRT-PCR in triplicate as described in [29], using primers targeting the E gene of SARS-CoV-2 (E_Sarbeco ACAGGTACGTTAATAGTTAATAGCGT; E_Sarbeco-R ATATTGCAGCAGTACGCACACA) (Sigma, France) and Luna Universal One-Step qRT-PCR Kit (New England Biolabs) on a Roche Light Cycler 480. The calibration of the assay was performed with a nCoV-E-Sarbeco-Control Plasmid (Eurofins Genomics, Germany) during PCR. However, the Ct values of the RT-qPCR in function of drug concentrations, normalized to the GAPDH housekeeping gene, were also assessed given the same results (see Supplemental Dataset, i.e., Supplemental Figures S1 and S2).

*2.6. Immunofluorescence and Confocal Microscopy Infected Cell Imaging*

A549-hACE2 cells seeded on glass coverslips were infected with SARS-CoV-2 at a MOI 0.01. At 45 h post-infection, cells were washed with PBS (Eurobio, France) and fixed in 4% paraformaldehyde in PBS for 15 min at room temperature, followed by permeabilization with 0.2% Triton X-100 (Sigma, Lezennes, France) in PBS for 4 to 5 min and blocking in 2% BSA in PBS for 15 min. Cell supernatants for each condition were used for virus quantification by RT-qPCR. Incubation with primary rabbit antibodies anti-SARS-CoV-2 membrane (M) protein (Rockland, MA, USA, 100-401-A55) (1:100) was performed for 2 h at room temperature. After washing with PBS, cells were incubated with secondary antibodies AF568-labeled goat-anti-rabbit (Invitrogen, Paris, France, A11011) (1:1000) for 2 h at room temperature. The coverslips were sealed with Prolong Diamond Antifade reagent (Fisher Scientific). Epifluorescence and confocal fluorescence images were generated using a Cell-Discoverer 7 LMS900 confocal laser-scanning microscope (Zeiss, Germany). DAPI-stained nuclei surfaces were analyzed using ZEN software (Otsu threshold method) from a grid of 8 × 8 images taken at 20× magnification. The percentage of polynuclear cells was quantified manually for a total of 1000–1200 cells. For intensity analysis, cells were imaged as Z stack with 0.3 µm sections, then Z projection images were processed for total intensity per cell using ImageJ/Fiji. Statistical tests were performed using Origin 8.5 software. Statistical significance was evaluated using one-way ANOVA tests, $p < 0.05$.

## 3. Results

### *3.1. In Vitro Antiviral Activity of Avermectins or Milbemycins in Simian Vero E6 Cells*

To determine if avermectins or milbemycins could protect cells from SARS-CoV-2 infection and to evaluate their toxicity, a cytopathic effect (CPE) reduction assay was first performed on African green monkey kidney-derived cell line Vero E6, which presents an ACE2 receptor and is widely used as an infection model for SARS-CoV-2 and for antiviral screening. We compared two commercial assays for determining cell viability, both based on quantitation of cellular metabolism, i.e., MTS and ToxGlo assays. Vero E6 cells were pretreated with serial dilutions of the drugs, then infected with SARS-CoV-2, and were kept in the medium with compounds for 72 h. Table 1 and Figure 1b recapitulate the main features of the evaluated compounds, including the dose–response curves for both assays and estimation of the 50% cytotoxic concentration (CC50).

**Table 1.** Antiviral activities of selected compounds against SARS-CoV-2 in Vero E6 cells, A549-hACE2 cells and Calu-3 cells. EC50 MTS, EC50 ToxGlo and EC50 RT-qPCR correspond to the 50% effective concentration determined by CPE reduction, using MTS or ToxGlo assay, or RT-qPCR, respectively. CC50: 50% cytotoxic concentration, SI: selectivity index.

| COMPOUND | EC50 ($\mu$M) MTS | EC50 ($\mu$M) ToxGlo | EC50 ($\mu$M) RT-qPCR | CC50 ($\mu$M) | SI |
|---|---|---|---|---|---|
| **VERO E6 cells** | | | | | |
| REMDESIVIR | 0.55 | 1 | | >50 | >49 |
| MOXIDECTIN | 5.3 | 3.5 | | 8.8 | 1.66 |
| DORAMECTIN | 2.4 | 1.1 | | 8.3 | 3.45 |
| SELAMECTIN | 3.9 | 2.3 | | 8.3 | 2.1 |
| IVERMECTIN | 3.6 | 3 | | 8 | 2.2 |
| MILBEMYCIN | 3.8 | 2.6 | | 7.8 | 2.05 |
| **A549 hACE2 cells** | | | | | |
| REMDESIVIR | 0.15 | | 0.31 | >10 | >30 |
| MOXIDECTIN | 1.3 | | 1.27 | 6.2 | 4.7 |
| IVERMECTIN | 2.1 | | 2.36 | 6.2 | 2.8 |
| MILBEMYCIN | 2.7 | | 2.8 | 6.1 | 2.2 |
| **Calu-3 cells** | | | | | |
| REMDESIVIR | | | 0.35 | >25 | >70 |
| MOXIDECTIN | | | 5.55 | 26 | 4.7 |
| IVERMECTIN | | | 3.36 | 22 | 5 |
| MILBEMYCIN | | | 6.78 | 23 | 3.3 |

As expected from previous studies [30,31], calculated EC50 values for our positive control, remdesivir, were in the low micromolar range (EC50 $_{REM\ MTS}$ = 0.55 $\mu$M and EC50 $_{REM\ TOXGLO}$ = 1.02 $\mu$M). Remdesivir showed a very favorable selectivity index (SI) (>50 with MTS, >100 with ToxGlo), in agreement with previous reports [30,31]. We found that avermectins and milbemycins protected infected cells from SARS-CoV-2-induced cell death in a dose-dependent manner (Figure 1b). Both MTS and ToxGlo assays provided similar results. The compounds, however, showed important toxicity, causing CC50 to drop to around 8 $\mu$M with SI, which hindered the numerical treatment of the dose–response curves that lacked a clear plateau for the highest concentration. In addition, cell survival curves showed a reproducible increase for all the tested molecules in the 1 to 5 $\mu$M range, before a dramatic drop in cell viability. Indeed, similar CC50 values have been reported for HeLa

cells treated with IVM, which has been proposed to induce cell cycle arrest, apoptosis and autophagy, presumably through mitochondrial pathways [32,33].

This effect was not seen upon treatment with remdesivir. Nevertheless, EC50 could be broadly estimated from the data and showed antiviral activity for IVM with EC50 $_{IVM\ MTS}$ = 3.6 µM and EC50 $_{IVM\ TOXGLO}$ = 3 µM, DOR with EC50 $_{DOR\ MTS}$ = 2.4 µM and EC50 $_{DOR\ TOXGLO}$ = 1.1 µM, and SEL with EC50 $_{SEL\ MTS}$ = 3.9 µM and EC50 $_{SEL\ TOXGLO}$ = 2.3 µM. MIL and MOX show antiviral activities within the same concentration range, with EC50 $_{MIL\ MTS}$ = 3.8 µM and EC50 $_{MIL\ TOXGLO}$ of 2.6 µM and EC50 $_{MOX\ MTS}$ = 5.3 µM and EC50 $_{MOX\ TOXGLO}$ = 3.5 µM, respectively.

### 3.2. In Vitro Antiviral Activity of IVM, MOX and MIL in Human Pulmonary Alveolar A549-hACE2 Cells

To assess the antiviral effect of IVM, MOX and MIL in a more relevant model, human pulmonary alveolar A549 cells, expressing the human ACE2 receptor thanks to a selected A549-hACE2 cellular clone upon transduction with a hACE2-containing lentiviral vector, were infected with SARS-CoV-2, following the same procedure as the Vero E6 cells. Cell viability/CPE was measured with the MTS assay. Additionally, viral RNAs were isolated from cell supernatant 72 h post-infection and quantified by RT-qPCR (Table 1, Figure S1). Cytotoxicity assays were performed in parallel in non-infected A549-hACE2 cells. The results are reported in Table 1 and Figure 2. Consistent with the data obtained with Vero E6 cells, we observed a strong dose-dependent reduction of infectivity and of the extracellular viral RNA levels in remdesivir-treated samples (EC50 $_{REM\ MTS}$ = 0.15 µM; EC50 $_{REM\ RTqPCR}$ = 0.31 µM), again associated with low cytotoxicity. The EC50 $_{A549-hACE2}$ for MOX, IVM and MIL were comparable both by MTS and by RT-qPCR and matched the EC50 measured in Vero E6 cells, in the range of 2 µM. Cellular toxicity was similar for the three tested compounds, close to ~6 µM. MOX appeared as the best compound with EC50 $_{MOX}$ = 1.3 and SI slightly increased, from 1.6 in Vero E6 cells to 4.7 in A549hACE2 cells. In addition, the increase in cell proliferation was not seen in the A549-hACE2 cells. Together, these results suggest that IVM, MOX and MIL provide inhibitory effects against SARS-CoV-2 infection in human pulmonary cells, albeit associated with poor selectivity.

### 3.3. In Vitro Antiviral Activity of IVM, MOX and MIL in Human Bronchial Calu-3 Cells

The cellular response to virus infection can differ depending on the infected cell type; therefore, we used the human bronchial epithelial Calu-3 cell lines, which express SARS-CoV-2 receptor ACE2 [34] and might better reflect physiological conditions than A549 cells ectopically expressing ACE2. In addition, Calu-3 cells, as with the airway epithelium, express low amounts of the cysteine protease cathepsin L (CTSL), which mediates SARS-CoV-2 entry into these cells by a pH-independent pathway. In contrast, entry of SARS-CoV-2 into Vero cells is CTSL-dependent and thus uses a pH-dependent pathway [35]. As SARS-CoV-2 infection does not produce evident CPE when infecting the Calu-3 cells, we monitored virus replication upon drug addition by quantifying intracellular viral RNA by RT-qPCR (Table 1, Figure S2). Toxicity of the compounds was evaluated by MTS assay. The results are reported in Figure 3 and Table 1. The control remdesivir was again found to be a potent SARS-CoV-2 inhibitor with EC50 = 0.35 µM and a SI greater than 70. Interestingly, cell viability was less affected upon incubation with IVM, MOX and MIL with CC50 estimated in the range of 22–26 µM. Consequentially, the antiviral activity of the three compounds could be defined more precisely and raised EC50 $_{MOX}$ = 5.55 µM (SI = 4.7), EC50 $_{IVM}$ = 3.36 µM (SI = 5) and EC50 $_{MIL}$ = 6.8µM (SI = 3.3). In summary, avermectins and milbemycins did efficiently block the infection of Calu-3 cells with SARS-CoV-2. The calculated EC50 were found in the same range than those found in Vero E6 and A549hACE2 cells, thus being cell-line independent, suggesting drug mechanistic effects that do not alter virus entry pathways [36].

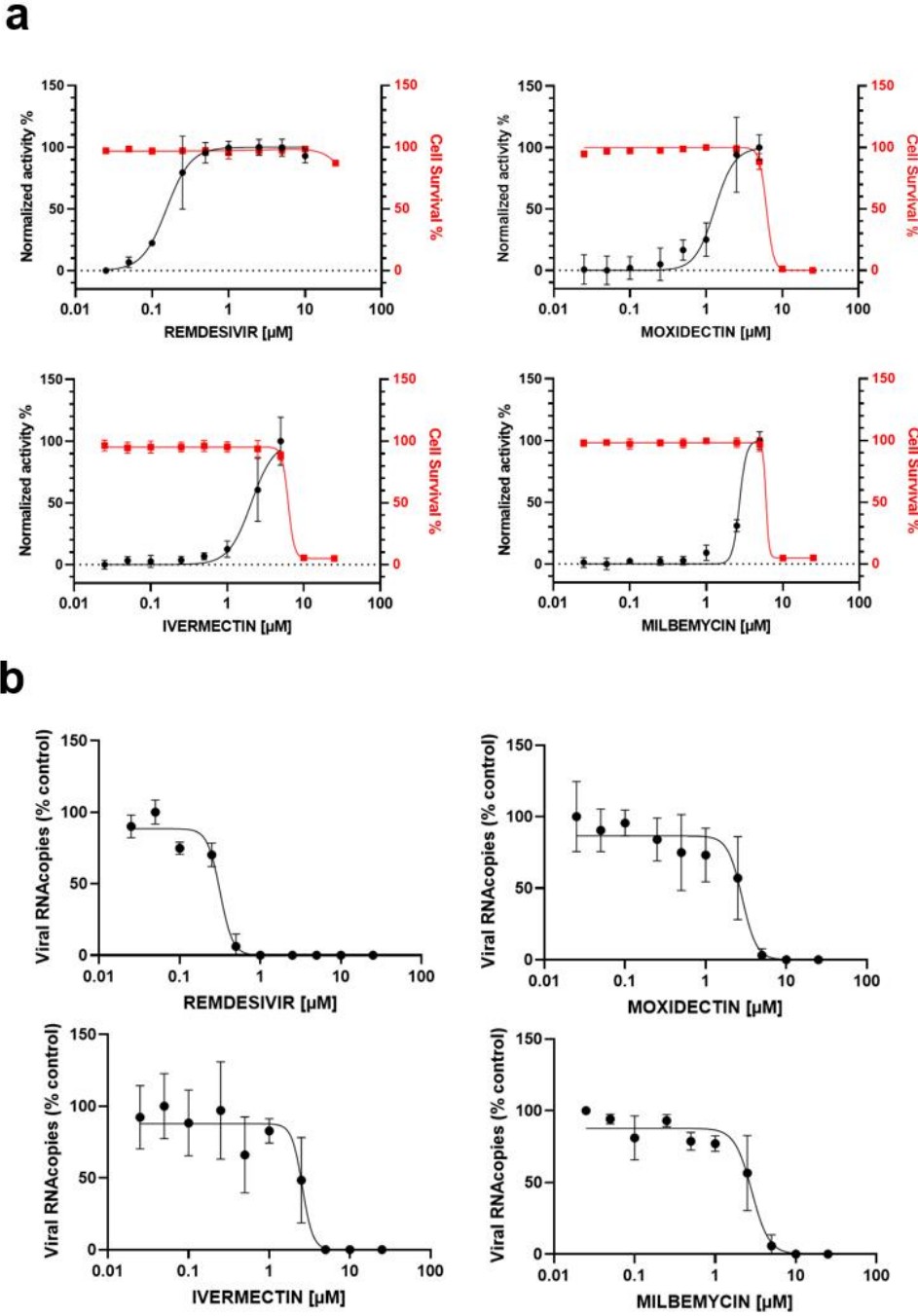

**Figure 2.** Antiviral and cytotoxic activities of selected compounds against SARS-CoV-2 infection in A549-hACE2 cells. A549-hACE2 were infected with SARS-CoV-2 at a MOI of 0.01 with 0.025, 0.05, 0.1, 0.25, 0.5, 1, 2.5, 5, 10 and 25 μM of the indicated compounds and incubated for 72 h. (**a**) Cytotoxic activity is shown in red curves and antiviral activity is shown in the black curves. The cytotoxicity of the drugs to A549-hACE2 cells was measured by MTS assay. Inhibition of virus replication was quantified by ToxGlo assay. The left and right *Y*-axes of the graphs stand for mean % inhibition vs. DMSO control (black) and mean % cell survival (red), respectively. The dashed lines are indicative of the baseline. (**b**) Viral RNA quantification by RT-qPCR from viral supernatant at 72 h. Data points are mean values and bars are the standard deviation of three independent experiments performed in duplicate.

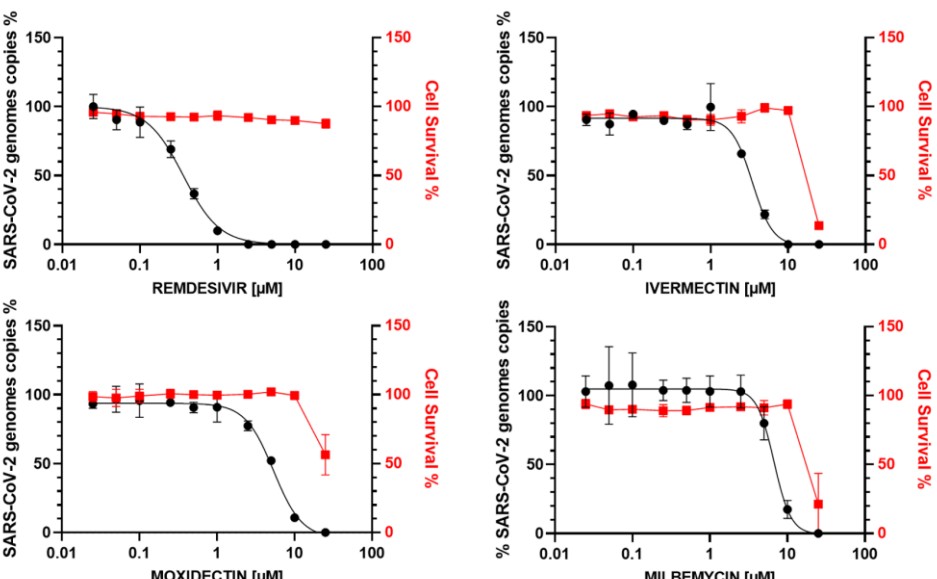

**Figure 3.** Antiviral and cytotoxic activities of selected compounds against SARS-CoV-2 infection in Calu-3 cells. Calu-3 cells were infected with SARS-CoV-2 at a MOI of 0.01 with 0.025, 0.05, 0.1, 0.25, 0.5, 1, 2.5, 5, 10 and 25 μM of the indicated compounds and incubated for 72 h. The cytotoxicity of the drugs to Calu-3 cells was measured by MTS assay. Inhibition of virus replication was quantified through viral RNA quantification by RT-qPCR from viral supernatant at 72 h. Cytotoxic activity is shown in red curves and the percentage of viral genome copies against the infected control is shown in the black curves. Data points are mean values and bars are the standard deviation of three independent experiments performed in duplicate.

*3.4. In Vitro Activity of IVM and MOX on SARS-CoV2 Shows a Reduction in Virus-Induced Cell Syncytia*

We finally confirmed MOX and IVM antiviral activities by *in cellulo* monitoring in A549 hACE2 cells after 45 h incubation in the presence of 4 μM of drugs (Figure 4a). The viral S/cellular ACE2 interaction has been shown to be responsible for the cell–cell fusion induced by SARS-CoV-2 infection [37]. As reported in Figure 4b, cell fusion was clearly visible in our assay using A549-hACE2 cells, where typical cells with multiple nuclei were detected using DAPI after infection with SARS-CoV-2. In contrast, non-significant cell fusion could be observed in non-infected cells (Figure 4c). We thus quantified the number of nuclei/cells (Figure 4c) or nucleus surface on a cell population in mock and infected cells and compared the results with infected cells treated with 4 μM IVM or 4 μM MOX. Both results unambiguously showed a strong limitation of cell fusions in drug-treated cells, confirming the properties of IVM and MOX to reduce viral dissemination in cell culture.

We also monitored the SARS-CoV-2 infection level by immunofluorescence labelling of SARS-CoV-2 M proteins. M is a transmembrane glycoprotein, one of the main structural protein of SARS-CoV-2 involved in particle formation and required for virion assembly [38–41]. M is found associated with intracellular compartments, where fluorescence of an anti-M antibody was detected on membranes of infected cells, and the endoplasmic Reticulum-related compartment (see Figure 4e). Quantification by confocal microscopy of M intensity per cell was used to compare virus production in infected and in IVM- or MOX-treated cells (Figure 4f), which showed a 2–3-fold decrease in the presence of MOX and IVM as compared to the wild-type. Overall, these results suggest that MOX and IVM interfere with virus induced cell syncytia and propagation in vitro.

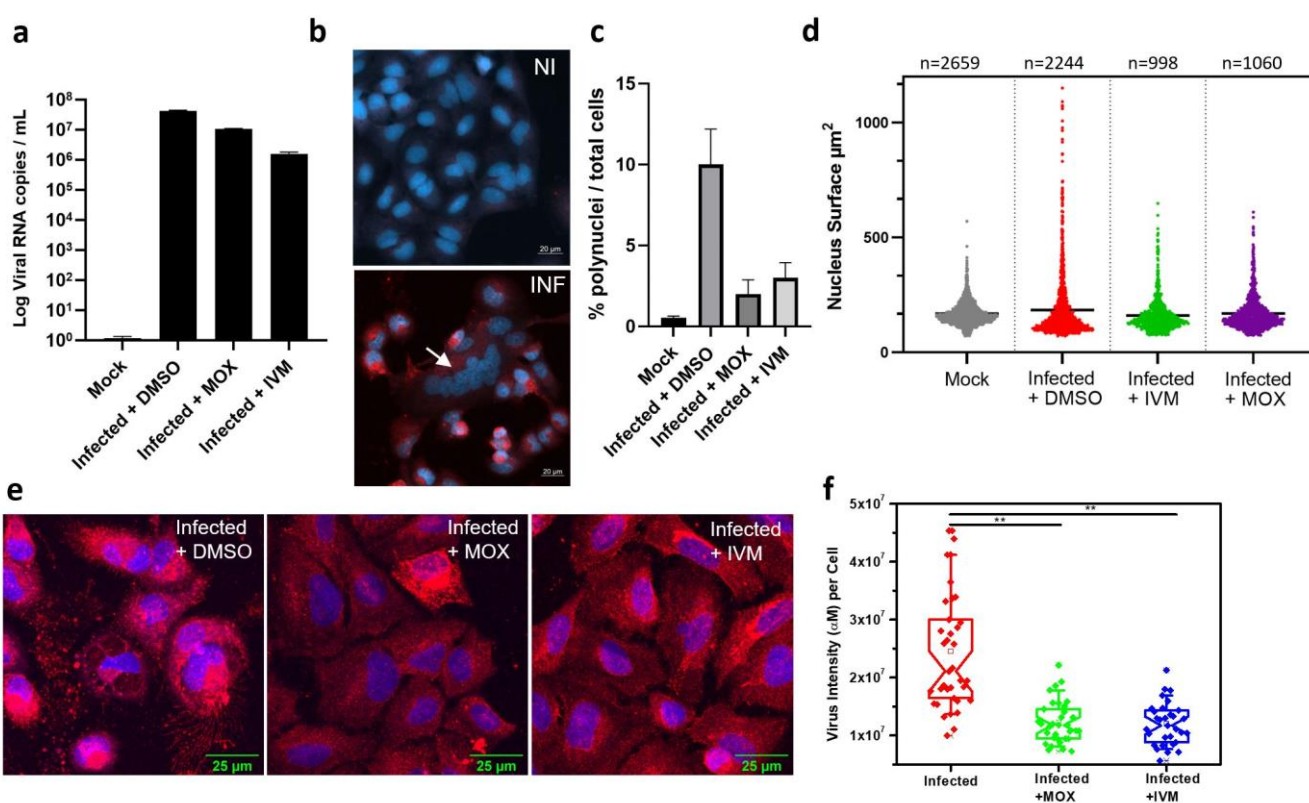

**Figure 4.** Cell monitoring and quantification of MOX and IVM antiviral activities. A549-hACE2 cells were infected with SARS-CoV-2 at MOI 0.01 in the presence of MOX (4 μM) or IVM (4 μM). A549-hACE2 cells were fixed at 48 h post infection, stained with DAPI and processed for immunofluorescence confocal microscopy using a SARS-CoV-2 membrane protein anti-M rabbit antibody and a secondary goat antibody anti-rabbit labelled with Alexa Fluor 568. (**a**) Viral RNA quantification from cell supernatant at 48 hpi. (**b**) Epifluorescence images of mock (NI) and infected cells (INF), the cells nuclei are shown in blue and the M protein in red. The arrow points towards giant cells containing multiple nuclei. The number of cell fusions observed in mock and in absence or presence of drugs in infected cells has been quantified by measuring the number of polynuclear cells (**c**) or the surface of nuclei in the cell population (**d**). In (**d**), *p* values are indicated and were computed by using two-sided independent t-tests and comparing the results to those for the untreated sample. (**e**,**f**) Changes in total intensity of virus (αM) per cell of SARS-CoV-2-infected A549-hACE2 cells with and without drugs IVM and MOX. (**e**) Images for intensity of virus (αM) with or without IVM and MOX 48 hpi. Scale bar is 25 μm. (**f**) Data for virus intensity per cell with or without drugs. A number of 40 < *n* < 50 cells were analyzed from at least 3 independent experiments. Statistical significant analysis were evaluated using one-way ANOVA tests and *t*-test. ** *p* < 0.01.

## 4. Discussion

Using SARS-CoV-2-infected A549-hACE2 cells, we evidenced EC50 measured by RT-qPCR of 1.3 μM, 2.4 μM and 2.8 μM for MOX, IVM and MIL, respectively. These results were consistent with those obtained using CPE in Vero E6 cells, with EC50 ranging from 2.4 μM (with MTS) or 1.1 μM (with ToxGlo) for DOR, to 5.3 μM (with MTS) or 3.5 μM (with ToxGlo) for MOX. These results were reproducible on bronchial epithelial cell culture (Calu-3), which is recognized as a classical model for studying human respiratory function [42].

We included IVM and remdesivir as controls for our screening procedure to compare our results to previously published data. For IVM, we found similar EC50 (2.1–2.36 μM on A549-hACE2 cells, 3–3.6 μM on Vero E6 cells, 3.36 μM on Calu-3 cells) to that reported by Caly et al. (2 μM). For remdesivir, we also found similar EC50 (0.15–0.31 μM) on A549-

hACE2 cells, 0.55–1 µM on Vero E6 cells, 0.35 µM on Calu-3 cells) to that reported by Wang et al. (0.77 µM).

Although IVM was thought to be a promising antiviral drug due to evidence of inhibition of SARS-CoV-2 replication in vitro, translating this activity in vivo is obviously subject to pharmacokinetics/pharmacodynamics issues, as recent apropos articles underlined. Repurposing drugs need to be accompanied by the evaluation of target plasma and lung concentrations following approved dosing in humans, for instance evaluating the lung $C_{max}$:EC50. We chose this approach to provide estimates comparable to those by Arshad et al., who reviewed the largest panel of candidate molecules [43], although one must keep in mind that it is overestimating the potency of drugs that require EC90% to be lower than $C_{min}$ to be truly qualifiable for further investigation.

In our study, MOX EC50 was found to range between 1.3 and 5.55 µM (equivalent to 832–3552 ng/mL or ng/g). As compared to IVM, higher $C_{max}$ values were generally observed for MOX, with studies showing $C_{max}$ 70.4 ng/mL following 400 µg/kg administration in horses [26] and as high as 234 ng/mL following 250 µg/kg administration in dogs [27]. Nonetheless, the observed $C_{max}$ varies widely across studies, with one study in sheep finding plasmatic $C_{max}$ of 28 ng/mL following administration of MOX 0.2 mg/kg orally [44]. The study by Lifschitz et al. is, to our knowledge, the only one documenting lung distribution of MOX. The subcutaneous administration of MOX 200 µg/kg to cattle (expected to lead to lower $C_{max}$ as compared to the oral route) gave plasma $C_{max}$ of 36 ng/g and lung $C_{max}$ of 63.7 ng/mL, indicating a possible lung to plasma ratio of around 2:1 for MOX distribution [45].

Cotreau et al. conducted the first safety study of MOX in humans as an ascending-dose study. In cohorts with higher doses, a slight increase in central neurological system grade 1–2 adverse events (nausea, vomiting and somnolence) was noticed; therefore, the enrollment was closed before the planed 5 mg cohort, given that pharmacokinetics is deemed the lowest posology for human use. Plasmatic $C_{max}$ was found to be 141 ng/mL in the 18 mg cohort (~250 µg/kg) and 289 ng/mL in the 36 mg cohort (~500 µg/kg). Administration to subjects fed with a high-fat meal is thought to enhance $C_{max}$ and AUC by >1/3 when compared to fasting subjects [46].

MOX has been FDA-approved for a single posology of 8 mg, even though it is usually used in veterinary medicine at the posology of 200 µg/kg. The recent phase 3 clinical trial [25] used the 8 mg posology for subjects weighing 50 kg on average, resulting in a posology of 160 µg/kg and showing a suitable tolerance profile on the largest cohort so far (*n* = 978 patients in the MOX group), as it had been shown in previous healthy volunteer studies [46–50].

Altogether, these data suggest that even an optimistic hypothesis (posology of 36 mg, lung to plasma ratio 2:1, high-fat feeding ×1.3) would still lead to expected lung $C_{max}$ < 750 ng/mg after administration of 5× the currently approved dose, for a predicted lung $C_{max}$/EC50 ratio < 0.2.

Very limited data on pharmacokinetics of other macrocyclic lactones (MIL, DOR and SEL) exist, mostly coming from isolated studies in animals.

SEL is exclusively used as a topical formulation of 6 mg/kg in cats and dogs, generally well tolerated [51]. Pharmacokinetic studies after IV (200 µg/kg) and oral (24 mg/kg) administration to cats/dogs showed plasma $C_{max}$ of 874/636 ng/mL and 11,929/7630 ng/mL, respectively, without raising any tolerance issue [52]. With an EC50 of 3.9 µM in our study (e.g., 3000 ng/mL), the hypothesized plasma $C_{max}$:EC50 $_{Vero-E6}$ for SEL at the posology of 24 mg/kg could therefore be close to 1.

DOR is used as an injectable form in ruminants, pigs and cattle, at the posology of 200 µg/kg and had equivalent plasma $C_{max}$ 32–52 ng/mL in cattle and horse as compared to IVM. Terminal $t_{1/2}$ was ~5 days in both studies [53,54]. With an EC50$_{Vero-E6}$ of 2.4 µM in our study (e.g., 2158 ng/mL), the hypothesized plasma $C_{max}$:EC50 $_{Vero-E6}$ for DOR at the posology of 200 µg/kg would therefore be ~0.02.

MIL is administered at the higher posology of 0.5 mg/kg in dogs and 2 mg/kg in cats for heartworm prevention. A mean $C_{max}$ of 353 ng/mL following 920 µg/kg administration was observed in dogs [55]. The lung to plasma ratio could also be around 2:1, as evidenced in rats in the original description of the drug by Ide et al. [56]. With an $EC50_{A549-hACE2}$ of 3.6 µM in our study (e.g., 1998 ng/mL), the hypothesized plasma $C_{max}:EC50_{A549-hACE2}$ for MIL at the posology of 920 µg/kg would therefore be ~0.34.

Along with encephalopathies following IVM administration in individuals with high *Loa loa* microfilaremia densities [57], the description of hypersensitivity reactions due to ABCB1 (formerly MDR1) mutations (encoding P-gp) following IVM administration has been recently evidenced [58]. This has been long known for animals, as those with the P-gp gene defect experience such adverse events for doses reaching 5 to 10 mg/kg for MIL and for DOR. SEL has a greater margin of safety than IVM in P-gp mutated dogs [59–61]. MOX is minimally metabolized and has low affinity for P-gp [62], which might convert into a lower rate of adverse events, but still requires cautious attitude when considering wide use.

Immunofluorescence imaging showed fewer infected cells upon IVM or MOX treatment, shown by the antibodies labelling (infected red cells), compared to the control, as well as the restoration of cell shapes, suggesting the interference of IVM/MOX on viral replication complexes, directly or indirectly. Infected MOX/IVM-treated cells showed an accumulation of the viral M proteins near or surrounding the nucleus (Figure 4). In addition, SARS-CoV-2-infected MOX/IVM-treated cells present a reduction of S-dependent syncytia in vitro (Figure 4), suggesting a prevention of syncytia formation by an unknown mechanism that could only be due to the reduction of viral replication. These results, although indirectly, are also compatible with a mechanism implying the ability of those drugs to bind to the importin (IMP) $\alpha$ hampering nuclear import [19,20], thus impairing viral replication in general or by interfering with the SpikeRBD-hACE2 complex formation, as suggested previously [63].

Our study is the first to examine antiviral parameters of macrocyclic lactones other than IVM on SARS-CoV-2; nevertheless, the data we provide here are only in vitro data, and the recent developments encourage us to emphasize that they should therefore be taken cautiously, and that many steps are to be completed before any result can be translated into clinical practice. Clinical risk/benefit balance is not in favor of the use of any of these molecules at this early stage. In order to try to approach the pharmacokinetic parameters of the molecules concerned, we used the data available for avermectins and milbemycins, often obtained from animal data, which do not directly allow an extrapolation to humans. Nevertheless, the concordance of all the data suggesting the impossibility of reaching even an imperfect pharmacological target of $C_{max}:EC50$ and therefore even less able to reach $C_{min}:EC90$ seemed important to us to avoid inappropriate conclusions being drawn from these data.

## 5. Conclusions

Altogether, our results clearly show that none of these agents are suitable for human use for their anti-SARS-CoV-2 activity per se. Our data show that DOR, MIL and SEL are not clinically relevant candidates. They are not approved for human use and therefore should not be used, and there is no rationale in our data to conduct the extensive studies that this would require (toxicology, impurity profile, thorough ADME evaluation).

Further steps could include in silico or in vivo studies, with the development of a pharmacokinetic model to stimulate lung exposure to MOX similarly to what was carried out by Jermain et al. [14] for IVM, or the translation of this research towards animal models of SARS-CoV2 infection to detect if there could paradoxically be some in vivo efficacy, even though pharmacokinetics profiles might not be as favorable as those observed for remdesivir, despite the latter showing no clear clinical benefit, at least in mortality [64]. An interesting option may also lie in other routes of administration, if tolerated, to enhance local lung delivery, as the proof-of-concept study on nebulized IVM by Chaccour et al. investigated [65]. However, another recent study also shows that in in vitro cell culture

of human primary airway epithelium, there is no inhibition of SARS-CoV2 replication at high doses of IVM or MOX, reinforcing the idea that this drug might have no effect on virus replication in vivo [66]. On the other hand, from the physio-pathological point of view of COVID-19, including the inflammatory variable to the disease, a very recent study reveals that IVM can moderate type I interferon responses and modulate several other inflammatory pathways due to SARS-CoV-2 infection, reducing secondary effects of the disease in infected hamster models [67]. Thus, this study supports the use of IVM as immunomodulatory drugs in the context of the late phases of SARS-CoV-2 infection but not by directly targeting the virus.

Finally, in the current context of new virus mutations [68], as well as the loss of post-vaccination immunization and the fact that some people cannot be vaccinated, the antiviral evaluation of old drugs as well as the development of new molecules is still of great interest, especially for the early phase of the disease, corresponding to the pre-inflammatory viral phase.

**Supplementary Materials:** The following are available online at https://www.mdpi.com/article/10.3390/covid2010005/s1, Figure S1: RT-qPCR, cycle threshold (Ct) from viral supernatant at 72 h in A549-hACE2 cells, Figure S2: RT-qPCR, cycle threshold (Ct) from viral supernatant at 72 h in Calu-3 cells.

**Author Contributions:** C.C.-B. set up the assays and performed infections and antiviral assays; M.H. and N.G. performed cell culture and produced viral stocks; P.M. established the A549-hACE2 cell line; C.C.-B. and S.L. performed data analysis and calculations; A.N. performed RT-qPCR and analysis; J.S. performed the immunofluorescence sample preparation and confocal microscopy acquisitions with S.L.; J.S. and S.L. performed image analysis; S.L., C.C.-B. and D.M. edited the figures; C.B., S.L., C.C.-B., C.C. and D.M. wrote the manuscript; C.B., A.M. and C.C. proposed the screening of avermectins and milbemycins family molecules; D.M. raised funding and directed the study with C.B. and C.C. All authors have read and agreed to the published version of the manuscript.

**Funding:** J.S. is a recipient of an Infectiopole Méditerranée fellowship. This study was supported by the CNRS and Montpellier University through a Montpellier Université d'Excellence (MUSE—COVID19M-FRS29-COVIDCEM) support, the IRD (UMI233 TransVIHMI), the REDSAIM Montpellier Metropole (2017–2021) project and the French Agency for Research (ANR COVID19—FRM Nucleo-CoV2 #159522).

**Institutional Review Board Statement:** Not applicable.

**Informed Consent Statement:** Not applicable.

**Data Availability Statement:** Not applicable.

**Acknowledgments:** We are grateful to Edouard Tuaillon and Vincent Foulongne for the provision of the SARS-CoV-2 strain from the Centre de Ressources Biologiques collection of the University Hospital of Montpellier, France, and to Monsef Benkirane (IGF, Montpellier, France) for providing the first amplification of this virus on Vero E6 cells. We are grateful to Inès Arfa for initial provision of MOX from Elanco Pharmaceutical Company. We also thank David Bracquemond for careful control check of the RT-qPCR data and figures of the supplemental data.

**Conflicts of Interest:** The authors declare no conflict of interest.

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
