# Peer review of "Low Selectivity Indices of Ivermectin and Macrocyclic Lactones on SARS-CoV-2 Replication In Vitro"

_covid, doi:10.3390/covid2010005_

Round 1

Reviewer 1 Report

Chable-Bessia and colleagues proposed a research article aimed at evaluating the antiviral efficacy of some drugs in COVID-19 in vitro models. The authors showed a very low selectivity index for the tested drugs (<10) suggesting how none of the tested agents is suitable for the treatment of SARS-CoV-2 infection due to the low selectivity index. Overall, the manuscript is interesting, however, there are some issues that the authors have to address before publication:
1)  Please remove the last sentence of the Abstract section;
2) Please indicate the catalog numbers of the antibodies used in chapter 2.6;
3) Have the authors tested the reduction of viral replication in the treated cells? Please better clarify this aspect ; 
4) I suggest to provide a unique Discussion section without dividing it into sub-chapters;
In the Discussion section, please add a brief description about the current and innovative diagnostic strategies used for the diagnosis of COVID-19 infection as well as the main symptoms associated with this infection. Also add information about vaccination and the need for testing the efficacy of antiviral therapies. For this purpose, please see:
- PMID: 33846767
- PMID: 34818922
- PMID: 32979398
- PMID: 34069418

Author Response

Montpellier, 12-7-2021

Dear editor,

Here are below the responses to the reviewers that we greatly thank for their comments and suggestions that have improved our manuscript.

All corrections are highlighted in yellow in the revised manuscript covid-1496088.

Best regards,

Delphine Muriaux

REVIEWER 1

Chable-Bessia and colleagues proposed a research article aimed at evaluating the antiviral efficacy of some drugs in COVID-19 in vitro models. The authors showed a very low selectivity index for the tested drugs (<10) suggesting how none of the tested agents is suitable for the treatment of SARS-CoV-2 infection due to the low selectivity index. Overall, the manuscript is interesting, however, there are some issues that the authors have to address before publication:

1)  Please remove the last sentence of the Abstract section;

  • This was done accordingly. The sentence « This is discuss in regards to recent clinical COVID studies on ivermectin » was removed from the abstract.

2) Please indicate the catalog numbers of the antibodies used in chapter 2.6;

  • Ok this was done accordingly. The catalogue number of were added section 2.3 page 4. A SARS-CoV-2 membrane protein anti-M rabbit antibody [Rockland, 100-401-A55] and a goat anti-Rabbit IgG secondary antibody Alexa Fluor 568 [A11011, Invitrogen].

3) Have the authors tested the reduction of viral replication in the treated cells? Please better clarify this aspect ; 

  • Indeed, yes, it is intrinsically in the results since the virus was not washed out during the course of the experiments (ie., replication curve in function of drug concentration): the experimental conditions are in the context of virus replication per se. The drugs were added 2h prior infection, then the cells were infected and neither the virus nor the drugs were washed out: the experiments are in the context of viral replication in treated cells.
  • This aspect was clarify by adding the sentence: In that context, the experiments were performed in an on-going viral replication in the treated cells. Page 4 section 2.4.

4) I suggest to provide a unique Discussion section without dividing it into sub-chapters;
In the Discussion section, please add a brief description about the current and innovative diagnostic strategies used for the diagnosis of COVID-19 infection as well as the main symptoms associated with this infection. Also add information about vaccination and the need for testing the efficacy of antiviral therapies. For this purpose, please see:
- PMID: 33846767
- PMID: 34818922
- PMID: 32979398
- PMID: 34069418

  • As suggested by the reviewer, we now provide a unique Discussion section.
  • As requested by the reviewer, we have implemented the introduction with the reference PMID: 32979398 [reference 8], page 2 lane 51: “ Therefore, the therapeutic arsenal is still limited and further options are needed [8] ».
  • Regarding the addition of a section on diagnostic strategies of the infection as well as the main symptoms, we believe that the proposed paper is based on the evaluation of macrocyclic lactones as potential therapeutic solutions and that many reviews have already dealt with these topics in a comprehensive way. Also, we believe that it is best to keep a single theme to this paper. However, we have added the following sentence to the discussion regarding the need to further develop and test the efficacy of antiviral therapeutics: "In the current context [66] of new virus mutations, as well as the loss of post-vaccination immunization, and the fact that some people cannot be vaccinated, the antiviral evaluation of old drugs as well as the development of new molecules is still of great interest, especially for the early phase of the disease, corresponding to the pre-inflammatory viral phase.» page 12 in the Conclusions section. Accordingly, the reference [66] (PMID: 34069418) was added.

Reviewer 2 Report

In the present study, the authors examined the in vitro antiviral activity of ivermectin against SARS-CoV-2. Two avermectins and two milbemycins were assessed in a clinical isolate of the virus from an infected patient in three different cell lines. The experiments are clearly described and designed for the intended experimental objectives. The results are clearly presented both in the text as well as the figures. The conclusions are supported by the results and do not support clinical development of these agents. This work is very important, given the attempts to repurpose such drugs as ivermectin and adverse clinical consequences. The authors acknowledge the limitations of their study; however, the results are an important contribution to the translation to clinical practice. I did not see reference to obtaining IRB approval and patient consent and kindly ask the authors to address it. The paper is well-written, organized and flows nicely. It was a pleasure to read.

Author Response

Montpellier, 12-7-2021

Dear editor,

Here are below the responses to the reviewers that we greatly thank for their comments and suggestions that have improved our manuscript.

All corrections are highlighted in yellow in the revised manuscript covid-1496088.

Best regards,

Delphine Muriaux

REVIEWER 2

In the present study, the authors examined the in vitro antiviral activity of ivermectin against SARS-CoV-2. Two avermectins and two milbemycins were assessed in a clinical isolate of the virus from an infected patient in three different cell lines. The experiments are clearly described and designed for the intended experimental objectives. The results are clearly presented both in the text as well as the figures. The conclusions are supported by the results and do not support clinical development of these agents. This work is very important, given the attempts to repurpose such drugs as ivermectin and adverse clinical consequences. The authors acknowledge the limitations of their study; however, the results are an important contribution to the translation to clinical practice. I did not see reference to obtaining IRB approval and patient consent and kindly ask the authors to address it. The paper is well-written, organized and flows nicely. It was a pleasure to read.

Response : we greatly thank the reviewers for the compliments and approbations.

As for the IRB approval and patient consent, it is referred in the section 2.2 page 3 of the manuscript : « SARS-CoV-2/CHU Montpellier/France was isolated from CPP Ile de France III, n°2020-A00935−34. This number contains all required approval at the “Centre de Ressources Biologiques” collection of the University Hospital of Montpellier (France)” which are the only reglementary information given by the CHU of Montpellier (France). This was included, page 3, in the section SARS-CoV-2 virus.

Round 2

Reviewer 1 Report

The authors partially answered to my previous comments No. 3 and No. 4. In particular, they should present data about the ct values observed through qRT-PCR during treatments, if available. 

As regards the Discussion section, I agree with the authors that the topic of their manuscript is on the effects of macrocyclic lactones as potential therapeutic solutions for COVID-19 patients. My previous comment was referred to a brief mention on the development of novel diagnostic, prognostic and therapeutic strategies that occurred in a relatively short time to effectively face the pandemic. 

Author Response

Dear reviewer,

Here are the responses to the request #3 and #4. I guess we did not completely understood the request #3 the first time.

  • In order to respond to this question more accurately, we added supplemental data information including figures S1 and S2 representing the Ct normalized to GAPDH in function of drug concentrations, showing the same conclusive results. A sentence was added in the section 2.5 page 4, lanes 175-177. Supplemental informations were uploaded.
  •  
  • For the comment on the Discussion, we are sorry to say that we have no opinion on novel diagnostic or prognostic that we feel are out of scope in this work. So we prefer not to add anything on vaccins. Thank you for your understanding.